# AI Pontryagin or how artificial neural networks learn to control dynamical systems

Lucas Böttcher 🄳 [1,2,4✉], Nino Antulov-Fantulin 🄳 [3✉] & Thomas Asikis[3,4✉]

The efficient control of complex dynamical systems has many applications in the natural and applied sciences. In most real-world control problems, both control energy and cost constraints play a significant role. Although such optimal control problems can be formulated within the framework of variational calculus, their solution for complex systems is often analytically and computationally intractable. To overcome this outstanding challenge, we present AI Pontryagin, a versatile control framework based on neural ordinary differential equations that automatically learns control signals that steer high-dimensional dynamical systems towards a desired target state within a specified time interval. We demonstrate the ability of AI Pontryagin to learn control signals that closely resemble those found by corresponding optimal control frameworks in terms of control energy and deviation from the desired target state. Our results suggest that AI Pontryagin is capable of solving a wide range of control and optimization problems, including those that are analytically intractable.

[1] Computational Social Science, Frankfurt School of Finance and Management, Frankfurt am Main 60322, Germany. [2] Department of Computational Medicine, University of California, Los Angeles, 90095-1766 Los Angeles, USA. [3] Computational Social Science, ETH Zurich, 8092 Zurich, Switzerland. [4] These authors contributed equally: Lucas Böttcher and Thomas Asikis. ✉email: l.boettcher@fs.de; anino@ethz.ch; asikist@ethz.ch

The problem of how to control complex systems has its roots in dynamical systems and optimization theory[1–4]. Mathematically, a dynamical system is said to be "controllable" if it can be steered from any initial state $\mathbf{x}_0$ to any target state $\mathbf{x}^*$ in finite time $T$. Controlling complex dynamical systems is relevant in many applications such as (i) development of efficient and robust near-term quantum devices[5,6], (ii) regulatory network control[7] in cellular biology, (iii) power-grid management[8], (iv) design of stable financial systems[9], and (v) epidemic management[10,11].

Historically, an early work by Kalman in the 1960s led to the formulation of an analytical condition for the controllability of linear systems based on the rank of the controllability matrix[1]. An equivalent condition, the so-called Popov–Belevitch–Hautus test[2], characterizes controllability of a linear system via its eigenmodes. More recently, concepts from the framework of structural controllability[3] have been used to control complex networks[12] with a minimum set of control inputs (i.e., driver nodes) that can be determined by identifying all unmatched nodes in a maximum matching problem. The direct application of this framework to general network controllability problems is, however, complicated by several factors[13]. First, finding the minimum set of driver nodes for an arbitrary network is NP hard[14]. Second, the design of an appropriate control signal is not specified in ref. [12] and its implementation may not be realizable in practice[13]. Third, in the presence of nodal self-dynamics, which were not considered in the controllability framework developed in ref. [12], a single time-varying input is sufficient to achieve structural controllability[15], challenging the findings of ref. [12].

Even if a certain control policy is able to steer a system towards a target state, it may not be possible to implement it in practice because of resource[16] and energy[17] constraints. To determine admissible control policies, one typically resorts to optimal control (OC) methods that rely on cost functions that one wishes to minimize. Such cost functions may be used to minimize the strength and frequency of control signals or, more generally, the "control energy"[17]. In technical networks, energy has to be supplied to control the action of underlying electrical and mechanical components. In social and socio-economic networks[18], one can identify control energy with the resources or costs incurred (e.g., economic and social costs of distancing policies) when changing the behavior of individual nodes[17]. Achieving optimal control of networked dynamical systems is thus a central task in control theory[19].

The solution of general optimal control problems is based on two main approaches: (i) Pontryagin's maximum principle[20,21] (necessary condition), which is a boundary-value problem in a Hamiltonian framework, or (ii) solving the Hamilton–Jacobi–Bellman (HJB) partial-differential equation (necessary and sufficient condition)[22]. Since the HJB equation usually does not admit smooth solutions[23], different approximate dynamic programming methods have been developed[24,25].

To extend the above approaches (i) and (ii) to complex and analytically intractable systems, different methods relying on artificial neural networks (ANNs) have been used to represent certain functions that appear in the formulation of optimal control problems. One possibility is to use ANNs to obtain an approximate solution to the value function of the HJB equation[24,26]. An alternative method is based on the solution of Pontryagin's maximum principle via differentiable programming[27]. Control approaches that rely on Pontryagin's maximum principle explicitly account for a control energy term in the loss function and are based on the solutions of a system's evolution and co-state equations. In addition to relying on an energy regularization term in the loss function, the control framework of ref. [27] is based on an extension of the maximum principle that includes higher-order derivatives, requiring the underlying dynamical systems to be twice-differentiable. Differentiable programming has been applied to control systems with a maximum of 13 state variables[27], almost two orders of magnitude smaller than some of the high-dimensional dynamical systems we study in this work.

Recent advances in automatic differentiation and physics-informed artificial neural networks[28] also contributed to the further development of modeling and control approaches. Physics-informed neural networks use Lagrangian and Hamiltonian-based formulations of physical models as priors for different learning tasks[29–31]. They are useful tools to model partially unknown systems[29] and have been also applied to control tasks[30,31].

Contrary to the above approaches, we show that it is possible to generate control signals that resemble those of optimal control[17] without relying on and solving maximum principle or HJB equations. To do so, we present AI Pontryagin, an ANN that overcomes several limitations of traditional optimal control methods resulting from the analytical and computational intractability of many complex and high-dimensional control tasks. AI Pontryagin extends neural ordinary differential equations (ODEs)[32] to general control problems, and efficiently steers complex dynamical systems towards desired target states by learning control trajectories that resemble those obtained with optimal control methods. It does so by exploring the vector field of the underlying dynamical system in a go-with-the-flow manner, without explicitly accounting for an energy-regularization term in the corresponding loss function. That is, AI Pontryagin minimizes the control energy[17] without evaluating an energy cost functional, leading to a substantially improved performance compared to existing control frameworks. Using analytical and numerical arguments, we show why AI Pontryagin is able to closely resemble the control energy of optimal control through an implicit energy regularization, resulting from the interplay of ANN initialization and an induced gradient descent. Further applications of such control methods to feedback control and a comparison with reinforcement learning are provided in ref. [33].

## Results

**Controlling dynamical systems with AI Pontryagin.** Before introducing the basic principles of AI Pontryagin, we first provide a mathematical formulation of the control problem of networked dynamical systems. We consider a network that consists of $N$ nodes and whose state is represented by the state vector $\mathbf{x}(t) \in \mathbb{R}^N$. Initially, nodes are in state $\mathbf{x}(0)$ and steered towards a target state $\mathbf{x}^*$ at time $T$ (i.e., $\mathbf{x}(T) = \mathbf{x}^*$) by means of suitable control inputs. Interactions between nodes are described by the dynamical system

$$\dot{\mathbf{x}}(t) = \mathbf{f}(\mathbf{x}(t), \mathbf{u}(t)) \tag{1}$$

and are subject to the constraint that the control function $\mathbf{u}(t) \in \mathbb{R}^M$ minimizes the cost functional

$$J = \int_0^T L(\mathbf{x}(t'), \mathbf{u}(t')) \; \mathrm{d}t' + C(\mathbf{x}(T)) \, . \tag{2}$$

The function $\mathbf{f} : \mathbb{R}^N \times \mathbb{R}^M \to \mathbb{R}^N$ in Eq. (1) accounts for both the interactions between nodes $1, \ldots, N$ and the influence of external control inputs $\mathbf{u}(t)$ on the dynamics. Note that the number of control inputs $M$ is smaller than or equal to $N$. For linear systems, we describe node-node interactions and external control inputs by $\mathbf{f}(\mathbf{x}, \mathbf{u}) = A\mathbf{x} + B\mathbf{u}$. The first term in Eq. (2) is the integrated cost over the control horizon $T$, e.g., the control energy

$$E_T[\mathbf{u}] = \int_0^T \|\mathbf{u}(t')\|_2^2 \; \mathrm{d}t' \tag{3}$$

if $L = \|\mathbf{u}(t')\|_2^2$. The quantity $C(\mathbf{x}(T))$ denotes the final cost (or

bequest value). Common formulations of optimal control include the control-energy term (3) directly in the cost functional[17,34]. In this case, minimizing the cost functional (2) corresponds to an explicit minimization of the control energy.

AI Pontryagin offers a complementary approach to reach a desired target state $\mathbf{x}^*$ in finite time $T$. To describe the basic principles of this control method, we proceed in two steps. First, we approximate and solve the dynamical system in terms of neural ODEs[32]. In particular, we describe the control input $\mathbf{u}(t)$ by an artificial neural network with weight vector $\mathbf{w}$ such that the corresponding control-input representation is $\hat{\mathbf{u}}(t; \mathbf{w})$. Second, we use a suitable loss function $J(\mathbf{x}, \mathbf{x}^*)$ and a gradient-descent algorithm to iteratively determine the weight vector $\mathbf{w}$ according to

$$\mathbf{w}^{(n+1)} = \mathbf{w}^{(n)} - \eta \nabla_{\mathbf{w}^{(n)}} J(\mathbf{x}, \mathbf{x}^*), \qquad (4)$$

where the superscript indicates the current number of gradient-descent steps, and $\eta$ is the learning rate. Unless otherwise stated, the loss function $J(\cdot)$ that we use in this work is the mean-squared error

$$J(\mathbf{x}(T), \mathbf{x}^*) = \frac{1}{N} \|\mathbf{x}(T) - \mathbf{x}^*\|_2^2. \qquad (5)$$

In order to calculate the gradients $\nabla_{\mathbf{w}^{(n)}} J(\cdot)$, we use automatic differentiation methods[35], where the gradients flow through an underlying artificial neural network that is time-unfolded[36] by ODE solvers[37]. We show a schematic of the forward and backward passes of AI Pontryagin and its coupling to a dynamical system in Fig. 1.

In the following paragraphs, we will show that AI Pontryagin approximates optimal control by minimizing the control energy (3), without including this term in the loss function (5) and without having any prior knowledge on the structure of optimal control signals. All neural-network architectures, hyperparameters, and numerical solvers are reported in the Methods.

**Approximating optimal control.** We now study the control performance of AI Pontryagin for linear systems (*i.e.*, $\mathbf{f}(\mathbf{x}, \mathbf{u}) = A\mathbf{x} + B\mathbf{u}$), for which there exist analytical OC inputs[17]

$$\mathbf{u}^*(t) = B^\top e^{A^\top (T-t)} W(T)^{-1} \mathbf{v}(T) \qquad (6)$$

that minimize the control energy $E_t[\mathbf{u}]$ [Eq. (3)]. For the derivation of Eq. (6), one applies Pontryagin's maximum principle to the Hamiltonian $H = \|\mathbf{u}(t)\|_2^2 + \boldsymbol{\lambda}(t)^\top [A\mathbf{x}(t) + B\mathbf{u}(t)]$[17], where $\boldsymbol{\lambda}(t)$ is an adjoint variable. The vector $\mathbf{v}(T) = \mathbf{x}(T) - e^{AT}\mathbf{x}_0$ in Eq. (6) is the difference between the target state $\mathbf{x}(T)$ and initial state $\mathbf{x}(0)$ under free evolution. The matrix $W(T)$ is the controllability Gramian and it is defined as

$$W(T) = \int_0^T e^{At} BB^\top e^{A^\top t} \, \mathrm{d}t. \qquad (7)$$

As an example of linear dynamics[17,34], we consider a two-state system with

$$A = \begin{pmatrix} 1 & 0 \\ 1 & 0 \end{pmatrix} \quad \text{and} \quad B = \begin{pmatrix} 1 \\ 0 \end{pmatrix}. \qquad (8)$$

The control task is to steer the system from $\mathbf{x}(0) = (1, 0.5)^\mathrm{T}$ to $\mathbf{x}^* = (0, 0)^\mathrm{T}$ in finite time $T = 1$.

In Fig. 2a, we show AI Pontryagin-controlled trajectories of the considered linear dynamics after 500 (blue), 1000 (purple), 1500 (red), and 30000 (orange) training epochs. The dashed black line represents a trajectory that we control with OC inputs (6). Note that the geodesic that connects $\mathbf{x}(0)$ and $\mathbf{x}^*(T)$ is not minimizing the control energy, because it would require large control inputs to steer the dynamics against the vector field (black arrows in

Fig. 2a). In alignment with the almost identical control trajectories of AI Pontryagin and OC, we also find that the energy evolution of AI Pontryagin almost perfectly coincides with that of OC (Fig. 2b), hinting at an implicit energy regularization of AI Pontryagin.

**Implicit energy regularization.** To provide insights into the observed implicit energy regularization of AI Pontryagin (Fig. 2b), we show that a gradient descent in the ANN weights $\mathbf{w}$ induces a gradient descent in the control input $\hat{\mathbf{u}}(t; \mathbf{w})$.

The evolution of the state vector $\mathbf{x}(t)$ is described by Eq. (1) and it is a function of $\hat{\mathbf{u}}(t; \mathbf{w})$. We now expand $\hat{\mathbf{u}}(t; \mathbf{w}^{(n+1)}) = \hat{\mathbf{u}}(t; \mathbf{w}^{(n)} + \Delta\mathbf{w}^{(n)})$ with $\Delta\mathbf{w}^{(n)} = -\eta \nabla_{\mathbf{w}^{(n)}} J$ for small $\Delta\mathbf{w}^{(n)}$ while keeping $t$ constant. This expansion yields

$$\hat{\mathbf{u}}(t; \mathbf{w}^{(n+1)}) = \hat{\mathbf{u}}(t; \mathbf{w}^{(n)}) + \mathcal{J}_{\hat{\mathbf{u}}} \Delta\mathbf{w}^{(n)}, \qquad (9)$$

where $\mathcal{J}_{\hat{\mathbf{u}}}$ is the Jacobian of $\hat{\mathbf{u}}$ with elements $(\mathcal{J}_{\hat{\mathbf{u}}})_{ij} = \partial\hat{u}_i/\partial\mathbf{w}_j$. Note that we can make $\Delta\mathbf{w}^{(n)}$ arbitrarily small by using a small learning rate $\eta$.

Since $\Delta\mathbf{w}^{(n)} = -\eta \nabla_{\mathbf{w}^{(n)}} J$ and $\nabla_{\mathbf{w}^{(n)}} J = \mathcal{J}_{\hat{\mathbf{u}}}^T \nabla_{\hat{\mathbf{u}}} J$, we obtain

$$\hat{\mathbf{u}}(t; \mathbf{w}^{(n+1)}) = \hat{\mathbf{u}}(t; \mathbf{w}^{(n)}) - \eta \mathcal{J}_{\hat{\mathbf{u}}} \mathcal{J}_{\hat{\mathbf{u}}}^T \nabla_{\hat{\mathbf{u}}} J. \qquad (10)$$

According to Eq. (10), a gradient descent in $\mathbf{w}$ [Eq. (4)] may induce a gradient descent in $\hat{\mathbf{u}}$, where the square matrix $\mathcal{J}_{\hat{\mathbf{u}}} \mathcal{J}_{\hat{\mathbf{u}}}^T$ acts as a linear transformation on $\nabla_{\hat{\mathbf{u}}} J$.

To better understand the implications of this result, we briefly summarize the control steps of AI Pontryagin. As described in the prior paragraphs and as illustrated in Fig. 2a, AI Pontryagin starts with a small initial control signal $\hat{\mathbf{u}}^{(0)}(t; \mathbf{w}^{(0)})$, then integrates the dynamical system (1), and performs a gradient descent in $\mathbf{w}$ according to Eq. (4). The closer the final state $\mathbf{x}(T)$ is to the target state $\mathbf{x}^*$, the smaller the loss (5) and the change in $\mathbf{w}$ [and in $\hat{\mathbf{u}}$ due to Eq. (10)]. If we initialize AI Pontryagin with a sufficiently small control input and learning rate, it will produce control trajectories that follow the vector field of the dynamical system in a go-with-the-flow manner and it will slowly adapt $\hat{\mathbf{u}}$ to reach the desired target state. Because of the induced gradient descent (10), the resulting control approximates OC methods that minimize the control energy (see the comparison between the final control energy of OC and AI Pontryagin in Fig. 2b, d). This way of controlling dynamical systems is markedly different from standard (optimal) control formulations[38] that are, for instance, based on Pontryagin's maximum principle and require one to explicitly minimize the control energy by (i) including an $\|\mathbf{u}\|_2^2$ term in the control Hamiltonian and (ii) solving an adjoint system[38]. AI Pontryagin thus provides a complementary approach for solving general control problems.

The induced gradient descent (10) can be directly observed in the positive correlations between $\|\Delta\mathbf{w}\|_2^2 = \|\mathbf{w}^{(n+1)} - \mathbf{w}^{(n)}\|_2^2$ and $\|\Delta\mathbf{u}\|_2^2 = \|\mathbf{u}^{(n+1)} - \mathbf{u}^{(n)}\|_2^2$ (Fig. 2c). Black disks indicate statistically significant correlation coefficients ($p < 10^{-9}$) that are each calculated for $10^3$ consecutive epochs and solid black lines are guides to the eye. After initializing AI Pontryagin for the linear two-state system (8) with weights that describe a small control input, we observe positive correlations between $\|\Delta\mathbf{w}\|_2^2$ and $\|\Delta\mathbf{u}\|_2^2$ with a large correlation coefficient of 0.96 for the first 1000 training epochs. The mean correlation coefficient is about 0.76. Changes in the correlation behavior reflect different training stages that are necessary to capture the strong curvature in the OC control trajectory (dashed black line in Fig. 2a). Between 1500 and 2000 training epochs, AI Pontryagin approximates the basic shape of the OC trajectory (solid red line in Fig. 2a) and then fine-tunes the weights $\mathbf{w}$ to match OC as closely as possible (solid orange line in Fig. 2a). The initial OC approximation phase that lasts up to about 2000 training epochs (before weight fine-tuning) is also visible in the evolution of $\|\mathbf{w}\|_2^2$ and $\|\mathbf{u}\|_2^2$ (Fig. 2d).

**a**  Dynamics evolving over time on a graph $G$:

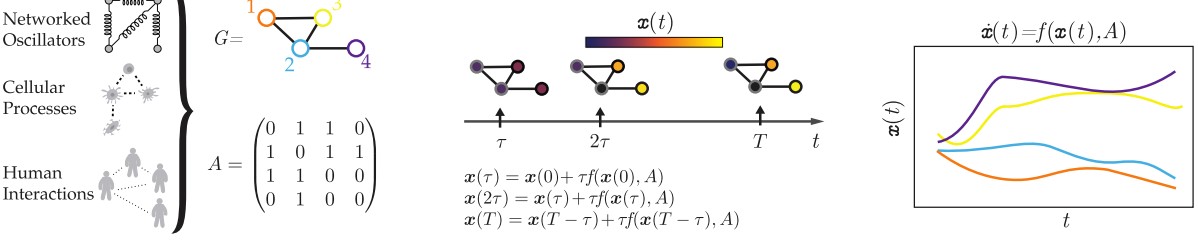

Complex dynamics represented by ordinary differential equations (ODEs)

Numerical solution of ODE system with discretization step $\tau$

Uncontrolled ODE dynamics

**b**  Controlling the evolution:

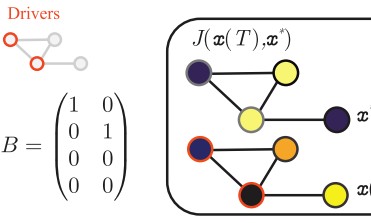  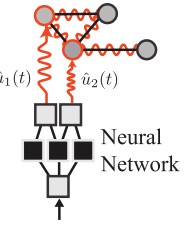  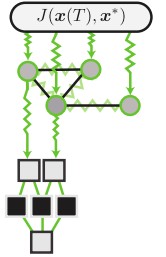  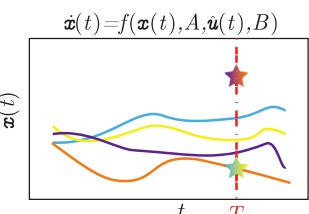

Matrix B encodes which nodes are controlled

Cost function $J(x(T), x^*)$ quantifies the difference between reached state $x(T)$ and target state $x^*$

Control function signal (orange) $\hat{u}(t)$ is generated with artifical neural network (ANN)

ANN parameters are updated (green) with gradient descent optimization w.r.t. cost function

Untrained ANN outputs control signals that drive the dynamics to undesired state $x(T)$ that is still far away from target $x^*$

**c**  Parameter optimization:

The ANN outputs a time-dependent control signal (orange)

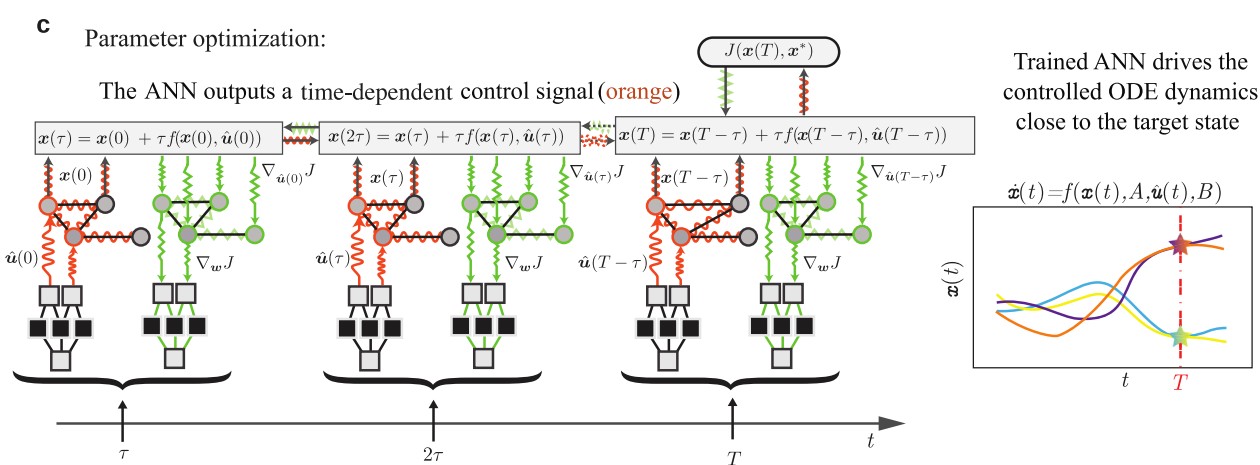

ANN parameters are updated (green) by gradient descent $-\eta \nabla J(x(T), x^*)$ over the unfolded ODE system

Trained ANN drives the controlled ODE dynamics close to the target state

**Fig. 1 Overview of AI Pontryagin. a** Illustration of a complex, uncontrolled dynamical system represented by ODEs, and discretization steps of a numerical solver. **b** Overview of basic elements necessary to control complex systems with ANNs. **c** Gradient-descent training of ANN parameters over the unfolded ODE system.

We again emphasize that the performance of AI Pontryagin and its induced gradient-descent mechanism depends on the choice of initial weights $\mathbf{w}^0$ (and thus on $\hat{\mathbf{u}}(t; \mathbf{w}^0)$). The initialization that we use to obtain the results shown in Fig. 2 is based on energy values that are small enough for AI Pontryagin to let it explore the vector field of the underlying dynamical system and approximate OC.

In the Supplemental Information (SI), we provide additional results that show that AI Pontryagin is able to control dynamics on directed networks. In particular, we show that AI Pontryagin can produce control signals with a control energy resembling that of the corresponding OC solution, which we verify by calculating

the corresponding optimal control signals if possible. In the SI, we also study the robustness of AI Pontryagin control with respect to different noise levels in the observed reached state.

After having outlined the mechanisms underlying the observed energy regularization of AI Pontryagin, we now turn towards non-linear systems.

**AI Pontryagin control of Kuramoto oscillators**. As an example of a non-linear system, we consider the Kuramoto model[39], which describes coupled oscillators with phases $\theta_i$ and intrinsic

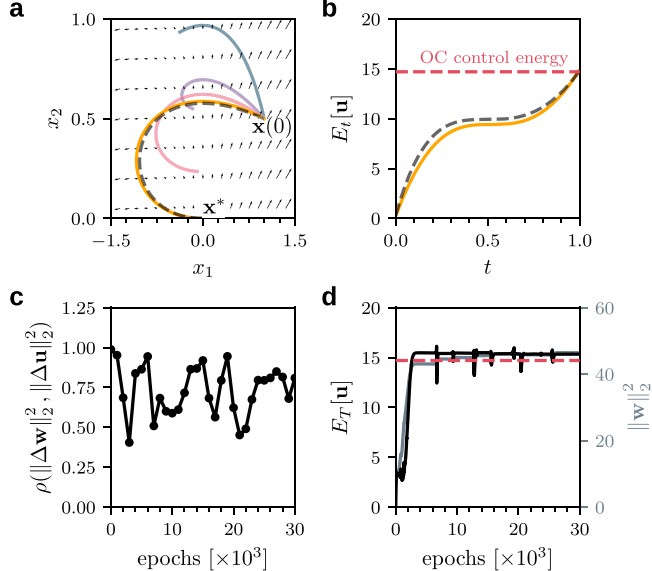

**Fig. 2 Controlling a two-node system with AI Pontryagin. a** Different AI Pontryagin-controlled trajectories of a linear system with $\mathbf{x}(0) = (1, 0.5)^{\top}$, $\mathbf{x}^* = (0, 0)^{\top}$, and $T = 1$ after 500 (blue), 1500 (purple), 2000 (red), and 30000 (orange) training epochs with learning rate $\eta = 0.02$. The dashed black line is the corresponding optimal control trajectory and black arrows indicate the vector field of the linear dynamical system $\mathbf{f}(\mathbf{x}, \mathbf{u}) = A\mathbf{x} + B\mathbf{u}$ with matrices $A$ and $B$ as in Eq. (8). **b** Evolution of the control energy $E_t[\mathbf{u}]$ of AI Pontryagin after 30000 training epochs (solid orange line) and of optimal control (dashed black line). **c** Correlations $\rho$ between squared norm differences of ANN weights $\mathbf{w}$ and control inputs $\mathbf{u}$. **d** The control energy $E_T[\mathbf{u}]$ (black solid line) and squared norm of ANN weights $\mathbf{w}$ (solid gray line) of AI Pontryagin as a function of training epochs. In panels (**b**, **d**), we indicate the total OC control energy by a dashed red line.

frequencies $\omega_i$ ($1 \leq i \leq N$) according to

$$\dot{\boldsymbol{\Theta}}(t) = \boldsymbol{\Omega} + \mathbf{f}(\boldsymbol{\Theta}(t), \mathbf{u}(t)),$$
$$\boldsymbol{\Theta}(0) = \boldsymbol{\Theta}_0, \tag{11}$$

where $\boldsymbol{\Theta} = (\theta_1, \dots, \theta_N)^{\top}$ and $\boldsymbol{\Omega} = (\omega_1, \dots, \omega_N)^{\top}$. In our following numerical experiments, we use natural frequencies and initial phases that are normally distributed with mean 0 and standard deviation 0.2. Interactions between oscillators and the influence of control inputs $u_i(t)$ on oscillator $i$ are modeled via

$$f_i(\boldsymbol{\Theta}(t), \mathbf{u}(t)) = \frac{K u_i(t)}{N} \sum_{j=1}^{N} A_{ij} \sin(\theta_j(t) - \theta_i(t)), \tag{12}$$

where $K$ is the coupling strength and $A_{ij}$ are the adjacency matrix components of the underlying (undirected) network. As a measure of synchronization at the final time $T$, we use the complete synchronization condition

$$|\dot{\theta}_i(T) - \dot{\theta}_j(T)| = 0 \text{ for } (i, j) \in E, \tag{13}$$

where $E$ is the set of edges[40,41]. If Eq. (13) is satisfied, all connected oscillators have constant phase differences. For control inputs that are equal to 1 (i.e., $u_i(t) = 1$ for all $i$), the oscillator system (11) has a unique and stable synchronized state if the coupling constant $K$ exceeds a critical value

$$K^* = \|L^{\dagger}\boldsymbol{\Omega}\|_{E,\infty}, \tag{14}$$

where $L^{\dagger}$ is the pseudo-inverse of the corresponding combinatorial graph Laplacian and $\|\mathbf{x}\|_{E,\infty} = \max_{(i,j) \in E} |x_i - x_j|$ is the maximum distance between elements in $\mathbf{x} = (x_1, \dots, x_N)^{\top}$ that are

connected by an edge in $E$[42]. In all numerical simulations, we use a subcritical coupling constant $K = 0.1 K^*$ such that control inputs $u_i(t) > 1$ are needed to synchronize the system.

For a global control $u(t)$ (i.e., $u_i(t) = u(t)$ for all $i$), there exists an OC input $u^*(t)$ satisfying

$$u^* = \min_u J(\boldsymbol{\Theta}(T), u) \tag{15}$$

$$J(\boldsymbol{\Theta}(T), u) = \frac{1}{2} \sum_{i,j} A_{ij} \sin^2(\theta_j(T) - \theta_i(T)) + \frac{\beta}{2} E_T[u], \tag{16}$$

where the parameter $\beta$ determines the influence of the energy regularization term on the cost function $J(\boldsymbol{\Theta}(T), u)$. Note that minimizing $J(\boldsymbol{\Theta}(T), u)$ is consistent with Eq. (13)[41].

An optimal control for the outlined non-linear control problem, the so-called adjoint-gradient method (AGM), can be derived using Pontryagin's maximum principle and a gradient descent in $u$[41]:

$$u^{(n+1)} = u^{(n)} - \tilde{\eta} \left[ \beta u^{(n)} + \frac{K}{N} \sum_{i=1}^{N} \lambda_i \sum_{j=1}^{N} A_{ij} \sin(\theta_j - \theta_i) \right], \tag{17}$$

where $\tilde{\eta}$ is the AGM learning rate and $\boldsymbol{\lambda} = (\lambda_1, \dots, \lambda_N)^{\top}$ is the solution of the adjoint system

$$-\dot{\lambda}_i = -\frac{K u \lambda_i}{N} \sum_{i \neq j} A_{ij} \cos(\theta_j - \theta_i)$$
$$+ \frac{K u}{N} \sum_{i \neq j} A_{ij} \lambda_j \cos(\theta_j - \theta_i), \tag{18}$$

with $\lambda_i(T) = 1/2 \sum_{i \neq j} A_{ij} \sin(2\theta_i(T) - 2\theta_j(T))$.

We compare the control performance of AI Pontryagin, which solves Eq. (11) using neural ODEs, with that of the AGM for a global control function. AI Pontryagin directly learns $\hat{u}(t; \mathbf{w})$ based on the following loss function without energy regularization term $\beta E_T[u]/2$:

$$J_1(\boldsymbol{\Theta}(T)) = \frac{1}{2} \sum_{i,j} A_{ij} \sin^2(\theta_j(T) - \theta_i(T)). \tag{19}$$

The learning rates $\eta$ [Eq. (4)] and $\tilde{\eta}$ [Eq. (17)] are chosen such that the ratio of the order parameter values of both control methods is approximately 1. We discuss in the SI that a high degree of synchronization can be achieved by controlling a fraction of all nodes and we show how a maximum matching approach[12] can be used to determine driver nodes for controlling linear dynamics with more than 1000 nodes. All employed network architectures and training parameters are summarized in the Methods and in our online code repository[43].

For a complete graph with $N = 225$ nodes and $T = 3$, we show the phase evolution of a system of uncontrolled oscillators with $u_i(t) = 1$ for all $i$ in Fig. 3a. As shown in Fig. 3b, AI Pontryagin can learn control inputs that drive the system of coupled oscillators into a synchronized state. A measure of the degree of synchronization is provided by the order parameter[39]

$$r(t) = N^{-1} \sqrt{\sum_{i,j} \cos[\theta_j(t) - \theta_i(t)]}. \tag{20}$$

Here, we used that the square of the magnitude of the complex order parameter $z = r e^{i\psi(t)} = N^{-1} \sum_{j=1}^{N} e^{i\theta_j(t)}$ can be expressed as

$$r(t)^2 = |z|^2 = N^{-2} \sum_{i,j} e^{i(\theta_j(t) - \theta_i(t))}$$
$$= N^{-2} \sum_{i,j} \cos[\theta_j(t) - \theta_i(t)].$$

A value of $r(t) = 1$ indicates that all oscillators have the same phase.

In Fig. 4 we show the evolution of the order parameter $r(t)$ and control energy $E_t[u]$ for both the AGM (solid lines) and AI

Pontryagin (dashed lines). We study the control performance of both methods on a complete graph (black lines), an Erdős–Rényi network $G(N, p)$ with $p = 0.3$ (blue lines), a square lattice without periodic boundary conditions (red lines), and a Watts–Strogatz network with degree $k = 5$ and a rewiring probability of 0.3 (green lines). All networks consist of $N = 225$ oscillators.

For all studied networks, we observe that AI Pontryagin reaches synchronization slightly faster than the AGM (Fig. 4a–d). We optimized the hyperparameters (e.g., the number of training epochs) of the artificial neural network underlying AI Pontryagin such that the control energy and degree of synchronization lie in a similar range to those of the AGM (Fig. 4e–h). Our results thus

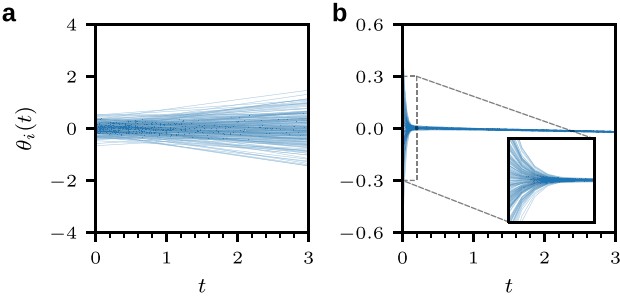

**Fig. 3 Synchronization of coupled oscillators.** The evolution of oscillator phases $\theta_i(t)$ ($1 \le i \le N$) in a complete network that consists of $N = 225$ coupled Kuramoto oscillators [Eqs. (11) and (12)] with a subcritical coupling constant $K = 0.1K^*$, which does not lead to synchronicity. All phases are initially distributed according to a normal distribution with mean 0 and standard deviation 0.2. **a** The control input is set to $u_i(t) = 1$ for all $i$ (uncontrolled dynamics), leading to increasing phase differences over time. **b** AI Pontryagin synchronizes the system of coupled oscillators.

indicate that AI Pontryagin is able to achieve control energies similar to those of OC also for non-linear networked dynamics.

Next, we discuss the ability of AI Pontryagin to control oscillator systems with a number of nodes that is about one order of magnitude larger than that considered in Fig. 4. We again compare AI Pontryagin with the AGM and set $T = 0.5$. Numerical experiments are performed on a square lattice (without periodic boundary conditions) that consists of $N = 2500$ coupled oscillators. We find that the control energy and order parameter ratios are $E_T^{\mathrm{AIP}}[u]/E_T^{\mathrm{AGM}}[u] \approx 1.0045$ and $r^{\mathrm{AIP}}(T)/r^{\mathrm{AGM}}(T) \approx 0.9999$, respectively. AI Pontryagin and the AGM reach similar order parameter and control energy values at time $T = 0.5$, indicating that both methods are able to control the larger-scale oscillator system.

For a runtime performance comparison, we also measure the learning time (or wall-clock time) associated with controlling the larger-scale oscillator system. To do so, we determine the runtimes of 50 AGM and 50 AI Pontryagin control realizations. The mean runtimes are 74s and 1.03s for the AGM and AI Pontryagin, respectively. For the studied oscillator system, the training time of AI Pontryagin is thus about two orders of magnitude smaller than that of the AGM. In the SI, we analyze the differences in runtime between AI Pontryagin and the AGM in more detail. To identify the main computational bottlenecks in the AGM, we performed a detailed runtime analysis of all code segments and found that the adjoint system solver requires very small step sizes to resolve the interaction between the adjoint system [Eq. (18)] and the gradient descent [Eq. (17)] in the control functions.

In a final numerical experiment, we show that AI Pontryagin is able to steer coupled Kuramoto oscillators to a target state that is different from the fully synchronized one. As an example of such a target state, we consider the control target to steer oscillators

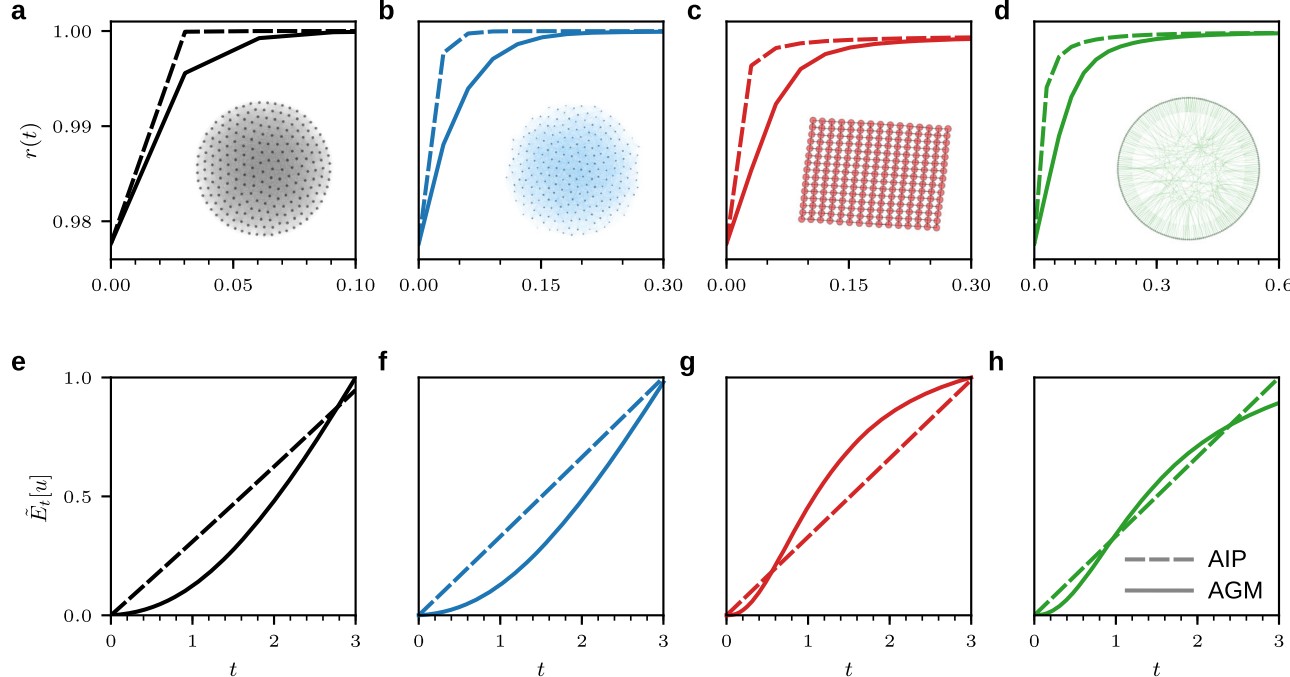

**Fig. 4 Controlling coupled oscillators with AI Pontryagin and the AGM.** We test the performance of AI Pontryagin and the AGM [Eqs. (17) and (18)] to control coupled Kuramoto oscillators [**a**, **e** complete network (black lines), **b**, **f** Erdős–Rényi network $G(N, p)$ with $p = 0.3$ (blue lines), **c**, **g** square lattice without periodic boundary conditions (red lines), and **d**, **h** Watts–Strogatz network with degree $k = 5$ and a rewiring probability of 0.3 (green lines)]. All graphs have $N = 225$ nodes and the total simulation time is $T = 3$. Panels (**a–d**) show the order parameter $r(t)$ and panels (**e–h**) show the normalized control energy $\tilde{E}_t[u] = E_t[u]/\max(E_t^{\mathrm{AIP}}[u], E_t^{\mathrm{AGM}}[u])$. The abbreviation AIP stands for AI Pontryagin. Dashed and solid lines indicate AI Pontryagin and AGM solutions, respectively.

## Oscillator phase evolution

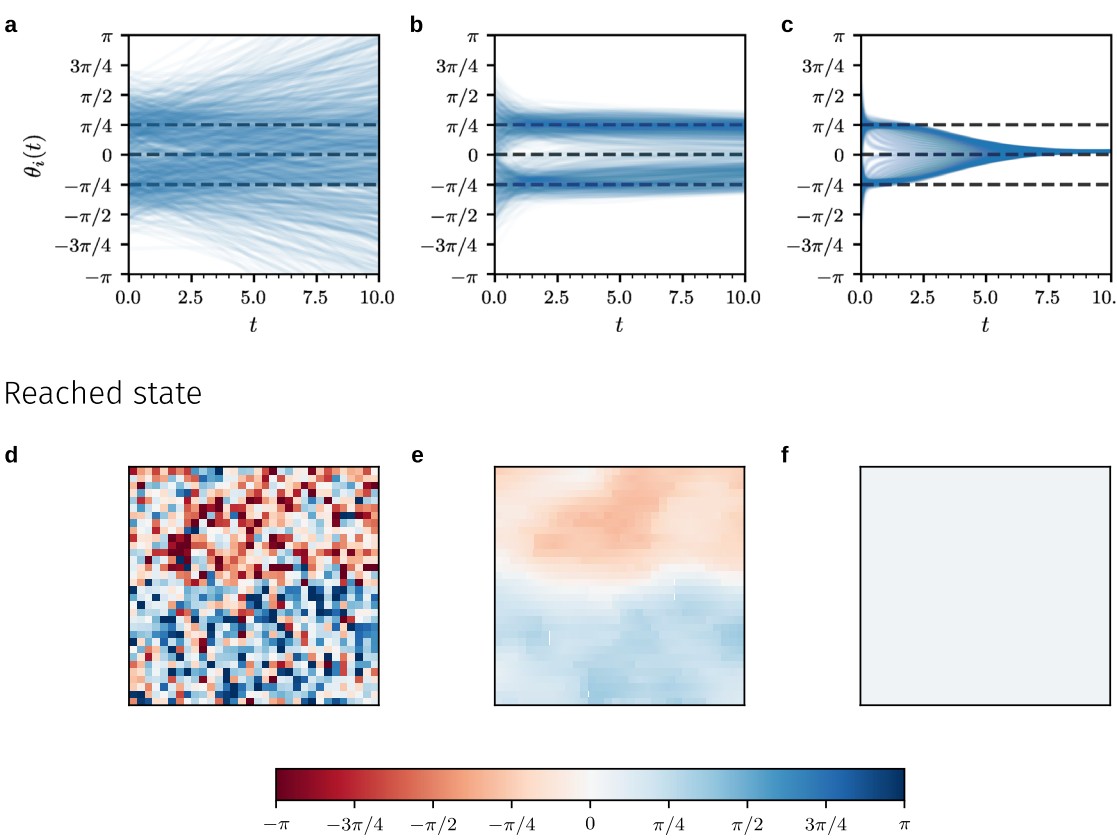

**Fig. 5 Kuramoto dynamics and different target states.** The evolution of the phases $\theta_i(t)$ ($1 \le i \le N$) of $N = 1024$ coupled Kuramoto oscillators [Eqs. (11) and (12)] that are arranged in a square lattice without periodic boundary conditions. We use a subcritical coupling constant $K = 0.01K^*$ and set $T = 10$. In all simulations, oscillator phases are initially distributed according to a bimodal Gaussian distribution with means $-\pi/4$ (top half of the lattice), $\pi/4$ (bottom half of the lattice), and variance 0.5. The top panels show the evolution of $\theta_i(t)$ and the bottom panels show the spatial distribution of oscillator phases $\theta_i(T)$ at time $T = 10$. Each pixel in the $32 \times 32$ bottom panels represents the phase value of one specific oscillator in the reached state. **a, d** The control input is set to $u_i(t) = 1$ for all $i$ (uncontrolled dynamics), leading to increasing phase differences over time. **b, e** AI Pontryagin steers oscillator phases towards $-\pi/4$ and $\pi/4$ by minimizing the loss function $J_2(\Theta(T))$ [Eq. (21)]. **c, f** AI Pontryagin synchronizes the system of coupled oscillators by minimizing the loss function $J_1(\Theta(T))$ [Eq. (19)]. The dashed black lines in panels (**a–c**) are guides-to-the-eye indicating phases with values $-\pi/4$, 0, and $\pi/4$. The learning rate is 15 in panels (**b, e**) and it is 0.12 in panels (**c, f**).

either towards $-\pi/4$ or $\pi/4$. This control goal can be described by the loss function

$$J_2(\Theta(T)) = \frac{1}{2} \sum_{i=1}^{N} \left[ \left| \theta_i(T) \right| - \frac{\pi}{4} \right]^2. \tag{21}$$

We show in Fig. 5 that AI Pontryagin can use the loss function (21) to reach target states in a lattice where $N = 1024$ coupled Kuramoto oscillators with subcritical coupling constant $K = 0.01K^*$ are arranged in two spatially separated groups, consisting of oscillators with phase values $\theta_i(T)$ ($1 \le i \le N$) that are approximately $-\pi/4$ or $\pi/4$, respectively. In Fig. 5a, d, we do not control the coupled Kuramoto oscillators, and we observe increasing phase differences over time. Using the loss function $J_2(\Theta(T))$ shows that AI Pontryagin can steer the system of coupled oscillators towards $-\pi/4$ and $\pi/4$ at the control time $T$ (Fig. 5b, e). Oscillators that are located in the upper half of the square lattice in Fig. 5e reached phase values close to $-\pi/4$ (indicated by light orange pixels), while oscillators in the lower half of the same lattice reached phase values of approximately $\pi/4$ (indicated by light blue pixels). By employing the loss function $J_1(\Theta(T))$, we can also use AI Pontryagin to synchronize the same system of coupled oscillators (dashed black line in Fig. 5c, f).

These results show that AI Pontryagin can be used in conjunction with different loss functions.

To summarize, AI Pontryagin has two key advantages over traditional adjoint-system-based control methods. First, approximate optimal control trajectories can be obtained without deriving and solving the adjoint system. The only inputs necessary are (i) a dynamical system, (ii) its initial state, and (iii) a desired target state. Second, the runtime of AI Pontryagin may be substantially faster than that of adjoint-gradient methods.

## Discussion

The optimal control of networked dynamical systems is associated with minimizing a certain cost functional, e.g., the strength and frequency of a control signal, or, more generally, the control energy (2). Traditional control approaches such as Pontryagin's maximum principle or the HJB equation are often analytically and computationally intractable when applied to complex dynamical systems.

In this work, we demonstrated the ability of AI Pontryagin, a control framework that is based on neural ODEs, to steer linear and non-linear networked dynamical systems into desired target states. AI Pontryagin uses as inputs the underlying dynamical

**Table 1 Learning rates $\eta$ (AI Pontryagin) and $\tilde{\eta}$ (AGM) used to control Kuramoto dynamics on different graphs. All ANNs use stochastic gradient descent for learning and only differ in their learning rate. The number of hidden layers and hidden layer neurons are 1 and 2, respectively. We use ELU activation functions and train the ANN for two epochs. At each node, we include a bias term and set all weights initially to a value of $10^{-3}$.**

| Graph | Edges (undirected) | Nodes | Learning rate $\eta$ | Learning rate $\tilde{\eta}$ |
|---|---|---|---|---|
| Complete | 25200 | 225 | 0.4 | 5 |
| Erdős–Rényi | 7569 | 225 | 0.4 | 5 |
| Square lattice | 420 | 225 | 0.32 | 25 |
| Watts–Strogatz | 450 | 225 | 0.31 | 15 |
| Square lattice | 4900 | 2500 | 0.0125 | 0.5 |

system and initial and target states. For the considered linear dynamics, we compared AI Pontryagin with corresponding analytical optimal control solutions and found that AI Pontryagin is not only able to drive undirected and directed complex networks of dynamical systems into desired target states, but also is able to automatically learn to approximate the optimal control energy. We supported this observation with analytical arguments and further compared AI Pontryagin with an optimal control method to evaluate the performance of both methods in synchronizing oscillators in different networks, again showing that AI Pontryagin is able to approximate the optimal control energy.

AI Pontryagin is a very versatile control framework that complements existing optimal control approaches and solves high-dimensional and analytically intractable control problems. Finally, there are various interesting avenues for further research. One possible direction for future work is the application of AI Pontryagin to solve complex quantum control problems to enhance robust performance of quantum systems[44]. Another possible direction for future research is to study the ability of AI Pontryagin to calculate optimal controls that preserve generator synchronicity during cascading failures, and ultimately avoid blackouts[45,46]. For such complex control tasks, it may be useful to combine physics-informed neural networks, such as those studied in ref. [29], with the proposed neural-network control approach to learn and control the dynamics of partially unknown systems.

## Methods

Both algorithms, AI Pontryagin and the AGM, are implemented in `PyTorch`.

All artificial neural networks that we use to represent the control input $\hat{\mathbf{u}}(t; \mathbf{w})$ in AI Pontryagin take the time $t$ as an input. To numerically integrate the studied dynamical systems, we apply the Dormand–Prince (DOPRI) method with adaptive step size during training and evaluation[47].

In the following paragraphs, we summarize the ANN architectures and hyperparameters that we used in our numerical experiments.

**Two-state system.** The artificial neural network that we use to control the two-state system (8) consists of a single hidden layer with 6 exponential linear units (ELUs). We transform the hidden layer output to the control signal via a linear layer with 1 neuron that describes the single control input in Eq. (8). We initialize the ANN weights $\mathbf{w}$ with the Kaiming uniform initialization algorithm[48]. For the gradient descent in $\mathbf{w}$ [Eq. (4)], we use the ADAM optimizer and set the learning rate $\eta = 0.02$. The number of time steps is 40.

**Kuramoto model.** The graph properties and ANN hyperparameters for controlling Kuramoto dynamics are summarized in Table 1. Independent of the underlying graph, we use the same number of hidden layers, hidden layer neurons, and training epochs for the numerical experiments that we performed to produce the results shown in Fig. 4. The activation function (ELU) is also the same in all numerical experiments. The number of time steps is 100. For the runtime comparison between AI Pontryagin and the AGM, we use the command `timeit` in `python`. In accordance with ref. [41], the energy regularization parameter $\beta$ of the AGM is set to $10^{-7}$ (see the SI for a more

detailed analysis of the AGM control performance on $\beta$). Initially, we set all ANN weights to a value of $10^{-3}$. Weight updates were performed using stochastic gradient descent. For the second numerical experiment involving the loss function (21), we use the Kaiman initialization method[48] and 16 ELU activations in one hidden layer.

## Data availability

Data supporting this study are publicly available at https://github.com/asikist/nnc.

## Code availability

All source codes and ANN architectures are publicly available at https://github.com/asikist/nnc.

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

## Acknowledgements

L.B. acknowledges financial support from the SNF (P2EZP2_191888). The work of N.A.F. was supported by the European Union - Horizon 2020 Program under the scheme "INFRAIA-01-2018-2019 - Integrating Activities for Advanced Communities", Grant Agreement no. 871042, "SoBigData++: European Integrated Infrastructure for Social Mining and Big Data Analytics" (http://www.sobigdata.eu). The authors thank Jan Nagler and both reviewers for their helpful comments.

## Author contributions

L.B., N.A.F., and T.A. wrote the manuscript and conceived the study. L.B., N.A.F., and T.A. developed models and analyzed data. L.B. and T.A. performed numerical simulations and contributed equally to this work.

## Funding

## Competing interests

The authors declare no competing interests.

## Additional information

**Peer review information** *Nature Communications* thanks Gang Yan and the other anonymous reviewer(s) for their contribution to the peer review this work. Peer reviewer reports are available.

