## [Peer Review File · Nature Communications]

Reviewers' Comments:

Reviewer #1:

Remarks to the Author:

When a networked dynamical system has a large size (i.e., many interacting nodes), the task of searching for an optimal input to control the system's behavior usually becomes computationally prohibitive. In the manuscript the authors proposed an ANN approach to overcome this challenge, by extending the recent framework of neural ordinary differential equations to optimal control. The most interesting part is that, although the loss function does not include energy cost explicitly, the (stochastic gradient descent based) training process of the ANN is able to naturally find an approximate solution of energy-minimization control.

The results are novel and interesting. However, I feel that the two cases in the current version used to demonstrate the approach's applicability are not convincing enough (especially when the word 'versatile' was used). In the following are several remarks which might be helpful for revision.

1. About desired state. Control usually means steering a system to any desired state. In the manuscript the authors showed that, by manipulating the coupling strengths between nodes the ANN approach can steer the Kuramoto network to a synchronized state. But, as we know, synchronization of this model is not that hard to achieve. To make the result more convincing, the authors need to try other final states, such as a fraction of the nodes are synchronized to one state and the other nodes to another (different) state. This could be doable by simply changing the objective described in Eq. (16).

2. About input nodes. The cool part of network control is that only a subset (not all) of nodes are required to receive the external control signals. In this manuscript the authors assumed that all nodes receive external signals (as described in Eq. (12)). I suggest at least testing the performance of this approach when only a proportion of nodes are driven, making a plot of the performance vs. the number (or proportion) of driver nodes.

3. Directed networks. The authors tested the approach on synthetic undirected networks. Actually, for directed networks the analytical treatment of optimal control would be indeed intractable, because when the adjacency matrix A is unsymmetrical the matrix exponential, i.e. $\exp(At)$ in Eq. (7), cannot be calculated via the eigen-decomposition trick. Hence, for directed networks (unless being very small), directly calculating the optimal inputs described by Eq. (6) is difficult. I am curious to know whether the ANN approach is able to solve this problem, i.e. whether it works for directed networks and with a subset of nodes being driven.

4. The writing can be improved. For instance, in Fig.2 caption "In (c,d), we indicate..." should be "In (b,d)..." ; Eqs. (17) and (21) are unnecessary as they are quite straightforward. Moreover, the authors have posted a related paper on arXiv (i.e. Ref. 25). If the arXiv paper is under review or has been published in another journal or conference, it might be necessary to explicitly state the difference between them and the innovation of the current manuscript.

In sum, the results are interesting but more cases are needed to demonstrate the approach's applicability. I will be happy to see a revised version.

Reviewer #2:

None

Reviewer #3:

Remarks to the Author:

The manuscript "AI Pontryagin or: How Artificial Neural Networks Learn to Control Dynamical Systems" presents a control framework to learn control signals that steer dynamical systems. The control approach is based on the function approximation of control inputs using a neural network, the parameters of which are trained in a form of neural ordinary differential equations (NODEs).

An interesting finding is that, without explicitly incorporating a control energy constraint, the proposed approach, AI Pontryagin, allows for learning of control signals that resembles those obtained with optimal control methods with an energy regularization term.

In the manuscript, the authors claim that AI Pontryagin is a very versatile control framework that complements existing optimal control approaches and is useful to solve high-dimensional and analytically intractable control problems. However, this is not the first neural-network-based approach for control problems. The similar approaches to approximate unknown control inputs and parameters with prior knowledge of dynamical systems have been intensively studied by many researchers. For instance, Roehrl et al. have proposed the similar control approach to model an external control force using a neural network [M. A. Roehrl et al. arXiv:2005.14617]. The training approach also seems to be similar with the authors' NODE-based training method. The NODE-based training has been applied to a real problem, which seems to be more advancing than the demonstrations presented in the manuscript.

Although AI Pontryagin assumes the prior knowledge on dynamical system Eq. (1), more general end-to-end learning framework to solve a broad class of learning and control tasks have been proposed [W. Jin et al. arXiv:1912.12970 (also in NeurIPS 2020)]. In my view, with comparison to previous neural-network-based control approaches, the conclusions that strengthen the importance of the proposed approach are missing.

In addition, I would suggest that the authors consider the following points:

- The authors state "AI Pontryagin is able to achieve control energies similar to those of OC also for nonlinear networked dynamics". I suspect that this statement is always true because the AGM is based on the cost function shown in Eq. (16), where the energy regularization term is introduced with a hyperparameter β . The energy constraint in the AGM can be adjusted with the hyperparameter; however, the authors state that the AI Pontryagin is similar to those of OC (trained by the AGM). Does this mean that the energy regularization of the AI Pontryagin depends on hyperparameters?

- The authors state "the training time of AI Pontryagin is about two orders of magnitude smaller than that of the AGM." This is an interesting advantage of the proposed approach, but it is unclear why the runtime (i.e., computational effort) is much smaller than the AGM. For the training, the authors also use the backpropagation-like approach shown in Fig. 1(c), which seems to be accompanied by a high computational effort, comparable to the adjoint gradient method in Eqs. (18) and (19). In my view, one of the claims is to ease the computational efforts to obtain the optimal solution and reduce the runtime. Thus, it will be important to more clearly explain this point.

- In my view, the robustness of optimal control is important when applying the proposed approach to a real-world control problem. However, the training method for control inputs relies on the derivatives (gradients) of the cost function and system's states, which may be sensitive to noise. It would be better if the authors discuss about the noise robustness of the proposed approach to strengthen the impact of this work.

- In Fig. 1(c): The update equation of $x(T)$ may be wrong. The authors should confirm it again.

- Page 3: "OC" may not be defined.

All in all, this manuscript is well written. The results supporting the present conclusions are appropriately shown. However, the lack of support for the above-raised points makes it unable to recommend publication of the manuscript in Nature Communications, at least in its present form.

Response to Reviewers: Manuscript ID NCOMMS-21-19781

October 11, 2021

Dear Editor and Reviewers,

Thank you for the careful consideration of our manuscript *AI Pontryagin or: How Artificial Neural Networks Learn to Control Dynamical Systems* and for the thoughtful comments and helpful suggestions.

In response to the comments of the reviewers, we have made appropriate changes in the manuscript to clarify it and to make it more comprehensive.

In addition to directly responding to the reviewers, we have also polished the language throughout the paper in our revision.

We believe that the revised manuscript is substantially improved and hope you will deem it ready for publication.

Sincerely,

Lucas Böttcher, Nino Antulov-Fantulin, and Thomas Asikis

Editor comments

Editor: Thank you again for submitting your manuscript “AI Pontryagin or: How Artificial Neural Networks Learn to Control Dynamical Systems” to Nature Communications. We have now received reports from 2 reviewers and, after careful consideration, we have decided to invite a major revision of the manuscript.

As you will see from the reports copied below, the reviewers raise important concerns. We find that these concerns limit the strength of the study, and therefore we ask you to address them with additional work. Without substantial revisions, we will be unlikely to send the paper back to review. In particular, the reviewers recommend to extend the analysis by considering directed networks and desired states beyond global synchrony; improve discussion on the amount of driver nodes, on computational cost and robustness of the proposed approach.

If you feel that you are able to comprehensively address the reviewers’ concerns, please provide a point-by-point response to these comments along with your revision. Please show all changes in the manuscript text file with track changes or colour highlighting. If you are unable to address specific reviewer requests or find any points invalid, please explain why in the point-by-point response.

Authors: We thank the editor for the detailed summary of the main points that were brought up by reviewers #1 and #3. We agree that the proposed extension of our analysis with an emphasis on directed networks, additional target states, driver-node selection, runtime, and noise sensitivity will contribute to a deepened understanding of the properties of the proposed neural-network-based control framework. All points are addressed in the point-by-point discussion below and all changes in the manuscript are shown in blue. We also added a detailed Supplemental Information (SI) file that provides in-depth discussions on the points raised by the reviewers.

Point-by-point discussion

Reviewer #1

Reviewer #1: When a networked dynamical system has a large size (i.e., many interacting nodes), the task of searching for an optimal input to control the system’s behavior usually becomes computationally prohibitive. In the manuscript the authors proposed an ANN approach to overcome this challenge, by extending the recent framework of neural ordinary differential equations to optimal control. The most interesting part is that, although the loss function does not include energy cost explicitly, the (stochastic gradient descent based) training process of the ANN is able to naturally find an approximate solution of energy-minimization control.

The results are novel and interesting. However, I feel that the two cases in the current version used to demonstrate the approach’s applicability are not convincing enough (especially when the word ‘versatile’ was used). In the following are several remarks which might be helpful for revision.

Authors: We thank reviewer #1 for their interest in our work and their very helpful and detailed suggestions for improvement. The outlined additional numerical experiments helped us to study different features and the versatility of the proposed control framework in more detail. All specific comments will be addressed below.

Reviewer #1: 1. About desired state. Control usually means steering a system to any desired state. In the manuscript the authors showed that, by manipulating the coupling strengths between nodes the ANN approach can steer the Kuramoto network to a synchronized state. But, as we know, synchronization of this model is not that hard to achieve. To make the result more convincing, the authors need to try other final states, such as a fraction of the nodes are synchronized to one state and the other nodes to another (different) state. This could be doable by simply changing the objective described in Eq. (16).

Authors: We agree with reviewer #1 that the study of the control performance for additional target states is helpful to assess the versatility of the proposed control method. As shown in Fig. 3 in the main text, the coupling constant in the original manuscript version was chosen such that synchronicity is not reached without an external control signal. Our results in Fig. 4 show that the proposed ANN method is able to synchronize such systems of coupled Kuramoto oscillators.

As suggested by reviewer #1, we complement these results by considering an additional target state, where a fraction of nodes are brought to one state and the remaining nodes to another state. We show in Fig. R1 that the presented ANN-based control framework can drive coupled oscillators to such a target state. A more detailed discussion of the corresponding numerical experiment can be found in the SI.

Actions taken:

- We added new numerical experiments in the revised manuscript on pages 7 and 8 to show that the proposed ANN-based control approach is able to steer coupled oscillators to a target state, in which a fraction of the nodes are synchronized to one state and the other nodes to another state.

Reviewer #1: 2. About input nodes. The cool part of network control is that only a subset (not all) of nodes are required to receive the external control signals. In this manuscript the authors assumed that all nodes receive external signals (as described in Eq. (12)). I suggest at least testing the performance of this approach when only a proportion of nodes are driven, making a plot of the performance vs. the number (or proportion) of driver nodes.

Authors: We thank reviewer #1 for this suggestion. To address this comment, we study the order parameter as a function of the proportion of nodes controlled. The results are shown in Fig. R2 (also added to the SI) and are aligned with those reported in [14], indicating that in oscillator networks with a large mean degree and a small coupling constant, a large proportion of nodes need to be controlled to achieve a certain degree of synchronization. The order parameter in networks with smaller mean degrees and a subcritical coupling constant is less sensitive to the fraction of controlled nodes.

Furthermore, in Fig. R3 we show that linear systems can be controlled with the ANN-based approach after determining the set of driver nodes according to the maximum matching method [8]. In the shown example, the proportion of driver nodes is about 50%.

We will discuss these results in the revised manuscript and include more detailed information in the SI.

Actions taken:

- We discuss the influence of different proportions of driver nodes on the control performance of our proposed framework in the revised manuscript (page 6) and in the SI (section III).

Oscillator phase evolution

Reached state

Figure R1: **Kuramoto dynamics and different target states.** The evolution of the phases $\theta_i(t)$ ($1 \leq i \leq N$) of $N = 1,024$ coupled Kuramoto oscillators [Eqs. (11) and (12) in the main text] that are arranged in a square lattice without periodic boundary conditions. We use a subcritical coupling constant $K = 0.01K^*$ and set $T = 10$. In all simulations, oscillator phases are initially distributed according to a bimodal Gaussian distribution with means $-\pi/4, \pi/4$ and variance 0.5. The top panels show the evolution of $\theta_i(t)$ and the bottom panels show the spatial distribution of oscillator phases $\theta_i(T)$ at time $T = 10$. Each pixel in the 32×32 bottom panels represents the phase value of one specific oscillator in the reached state. **(a,d)** The control input is set to $u_i(t) = 1$ for all i (“uncontrolled dynamics”), leading to increasing phase differences over time. **(b,e)** AI Pontryagin steers oscillator phases towards $-\pi/4$ and $\pi/4$ by minimizing the loss function $J_2(\Theta(T))$ [Eq. (21) in the main text]. **(c,f)** AI Pontryagin synchronizes the system of coupled oscillators by minimizing the loss function $J_1(\Theta(T))$ [Eq. (19) in the main text]. The dashed black lines in (a–c) are guides-to-the-eye indicating phases with values $-\pi/4, 0$, and $\pi/4$. Learning rates varied between 10^{-2} – 10^1 in (b,e) and 0.1 in (c,f).

Figure R2: **Dependence of the order parameter on the fraction of driver nodes.** The order parameter $r(T)$ [Eq. (20) in the main text] of Kuramoto oscillator dynamics as a function of the fraction of controlled nodes for **(a)** an Erdős–Rényi network $G(N, p)$ with $p = 0.3$ and **(b)** a Watts–Strogatz network with degree $k = 5$ and rewiring probability $p = 0.3$. Both graphs have $N = 225$ nodes. Hence, the mean degree of the Erdős–Rényi network is 67.5. We set $T = 5$ and used learning rates η between 0.1 and 1.0. The shown results are based on averages over 1000 i.i.d. natural frequency realizations. Different markers indicate different coupling constants: $K = 0.1K^*$ (blue disks), $K = 0.15K^*$ (orange squares), and $K = 0.2K^*$ (green triangles).

Figure R3: **Control of linear systems.** **(a)** Final loss [Eq. (5) in the main text] and control energy [Eq. (6) in the main text] as a function of training epochs. **(b)** Total control energy of AI Pontryagin and optimal control [Eq. (6) in the main text]. The fraction of controlled nodes in (a,b) is 50%. We determined driver matrix B and driver nodes according to the maximum matching method [8].

Reviewer #1: 3. Directed networks. The authors tested the approach on synthetic undirected networks. Actually, for directed networks the analytical treatment of optimal control would be indeed intractable, because when the adjacency matrix A is unsymmetrical the matrix exponential, i.e. $\exp(At)$ in Eq. (7), cannot be calculated via the eigen-decomposition trick. Hence, for directed networks (unless being very small), directly calculating the optimal inputs described by Eq. (6) is difficult. I am curious to know whether the ANN approach is able to solve this problem, i.e. whether it works for directed networks and with a subset of nodes being driven.

Authors: We thank reviewer #1 for proposing to study the ability of the ANN control approach to learn control signals for directed networks. To be able to compare the ANN approach with the optimal solution [Eq. (7) in the revised manuscript], we study a directed 3-node cogwheel system with (i) an antisymmetric and invertible adjacency matrix (see Fig. R4), and (ii) an antisymmetric and non-invertible adjacency matrix (see Fig. R5). In both examples, one driver node is selected. We discuss in the SI that the ANN approach is still able to produce control signals with a control energy that resembles that of the corresponding optimal control (OC) solution, which we verify with analytical calculations for both systems. We also study a larger-scale directed network (see Fig. R6) for which the **OC solution cannot be calculated** if the fraction of controlled nodes is too small, because the corresponding controllability Gramian is not invertible (as suggested by the reviewer). We find that the ANN method is still able to minimize the loss function and steer the dynamics to a state that is very close to the target state. More details on these results are reported in the SI.

Figure R4: **Controlling a connected 3-node cogwheel system.** (a) Evolution of system states $x_i(t)$ ($1 \leq i \leq 3$) under OC (dashed lines) and neural-network-based controls (solid lines). (b) Evolution of the control energy $E_t[\mathbf{u}]$ for AI Pontryagin after 20,000 training epochs (solid orange line) and optimal control (dashed black line). The learning rate is $\eta = 6 \times 10^{-4}$.

Actions taken:

- In the SI (section I) we demonstrate the ability of the proposed ANN approach to control directed networks and produce control signals with a control energy that resembles that of OC solutions (if available). We also discuss an example for which OC signals cannot be calculated. We show that the ANN approach still steers the dynamics to a state that is very close to the desired target state.
- We mention these new numerical experiments in the revised manuscript: “In the Supplemental Information (SI), we provide additional results on the ability of AI Pontryagin to control dynamics on directed networks. We show that AI Pontryagin is able to produce control signals with a control energy resembling that of the corresponding OC solution, which we verify by calculating the corresponding optimal control signals if possible.”

Figure R5: **Controlling an unconnected 3-node cogwheel system.** (a) Evolution of system states $x_i(t)$ ($1 \leq i \leq 3$) under OC (dashed lines) and neural-network-based controls (solid lines). (b) Evolution of the control energy $E_t[\mathbf{u}]$ for AI Pontryagin after 20,000 training epochs (solid orange line) and optimal control (dashed black line). The learning rate is $\eta = 6 \times 10^{-4}$.

Figure R6: **Controlling a larger-scale directed network.** We use AI Pontryagin to control a growing network [7] (directed graph) with 1,024 nodes for two different fractions of controlled nodes that are selected uniformly at random. (a) If 95% of all nodes are controlled, the controllability Gramian in Eq. (7) is non-invertible and OC solutions (6) (see main text) cannot be calculated. AI Pontryagin still manages to steer the dynamics towards the target state and minimize the final loss $J(\mathbf{x}(T), \mathbf{x}^*)$. (b) For 99% controlled nodes, both methods minimize the loss function. Dashed and solid lines indicate AI Pontryagin and OC solutions, respectively.

Reviewer #1: 4. The writing can be improved. For instance, in Fig. 2 caption “In (c,d), we indicate...” should be “In (b,d)...”; Eqs. (17) and (21) are unnecessary as they are quite straightforward. Moreover, the authors have posted a related paper on arXiv (i.e., Ref. 25). If the arXiv paper is under review or has been published in another journal or conference, it might be necessary to explicitly state the difference between them and the innovation of the current manuscript.

Authors: We thank reviewer #1 for pointing out these typos and nonessential equations. In the revised manuscript, we corrected the typo in the caption of Fig. 2 and removed Eq. (17), and moved Eq. (21) to a footnote. We used the cited arXiv preprint as a summary of results that are related to ANN-based control methods. We developed the original version of the cited arXiv preprint into two separate works. The work we submitted here focuses on ANN-based control of complex dynamical systems, their implicit control energy regularization properties, and the connection to optimal control methods that rely on Pontryagin’s maximum principle. In our separate work (that is still not published), we evaluate an altered version of the presented ANN-based control methods, which explicitly focuses on feedback control tasks and relies on other NN architectures (e.g., graph neural networks), and we compare these feedback-control methods with reinforcement learning techniques that are relevant to approximate HJB schemes. In the revised introduction, we provide a more detailed discussion of related work and our contributions.

Actions taken:

- We corrected the typo “(c,d)” in the revised caption of Fig. 2.
- We removed Eq. (17) and moved Eq. (21) to a footnote.
- In the revised introduction, we provide a more detailed discussion of the connection of our work to related work and its contributions.

Reviewer #1: In sum, the results are interesting but more cases are needed to demonstrate the approach’s applicability. I will be happy to see a revised version.

Authors: We again thank reviewer #1 for the very helpful and constructive comments that made our revised paper more stronger. In the revised manuscript and SI, we provide more details on different target states, the influence of different proportions of driver nodes, and control of dynamical systems on directed networks.

Reviewer #3

Reviewer #3: The manuscript “AI Pontryagin or: How Artificial Neural Networks Learn to Control Dynamical Systems” presents a control framework to learn control signals that steer dynamical systems. The control approach is based on the function approximation of control inputs using a neural network, the parameters of which are trained in a form of neural ordinary differential equations (NODEs). An interesting finding is that, without explicitly incorporating a control energy constraint, the proposed approach, AI Pontryagin, allows for learning of control signals that resembles those obtained with optimal control methods with an energy regularization term.

Authors: We thank reviewer #3 for the positive feedback and their time in carefully reviewing our manuscript. We address all specific comments below.

Reviewer #3: (1) In the manuscript, the authors claim that AI Pontryagin is a very versatile control framework that complements existing optimal control approaches and is useful to solve high-dimensional and analytically intractable control problems. However, this is not the first neural-network-based approach for control problems. The similar approaches to approximate unknown control inputs and parameters with prior knowledge of dynamical systems have been intensively studied by many researchers. For instance, Roehrl et al. have proposed the similar control approach to model an external control force using a neural network [M. A. Roehrl et al. arXiv:2005.14617]. The training approach also seems to be similar with the authors' NODE-based training method. The NODE-based training has been applied to a real problem, which seems to be more advancing than the demonstrations presented in the manuscript. Although AI Pontryagin assumes the prior knowledge on dynamical system Eq. (1), more general end-to-end learning framework to solve a broad class of learning and control tasks have been proposed [W. Jin et al. arXiv:1912.12970 (also in NeurIPS 2020)]. In my view, with comparison to previous neural-network-based control approaches, the conclusions that strengthen the importance of the proposed approach are missing.

Authors: We thank reviewer #3 for the detailed feedback on related works.

Very **briefly**, in contrast to the work of Jin et al. (2020), Roehrl et al. (2020), and other control approaches, the proposed control framework is able to efficiently produce control signals with the following distinct properties:

- effective control of **high-dimensional** dynamical systems (few orders of magnitude larger than in the mentioned references),
- learning to control **complex networks** of dynamical systems, including systems with **directed** networks.
- generation of control signals that resemble those of optimal control *without* solving Pontryagin's maximum principle (PMP) or the Hamilton–Jacobi–Bellman (HJB) equation due to **implicit energy regularization**.

In addition to the above summary, we also add a more **detailed discussion** on the comments of reviewer #3 below.

Recent advances in differentiable programming (*i.e.*, automatic differentiation through computational graphs) and physics-informed neural networks [Karniadakis et al. Physics-informed machine learning. Nat. Rev. Phys. 2021] led to several developments in control theory. In accordance with the distinction between Lagrangian and Hamiltonian formulations of dynamical systems in classical mechanics, physics-informed neural networks can be also divided into these two subclasses:

- Lagrangian-based formulation [Roehrl et al. (2020). Modeling System Dynamics with Physics-Informed Neural Networks Based on Lagrangian Mechanics], [Lutter et al. (2019). Deep Lagrangian Networks: Using Physics as Model Prior for Deep Learning]
- Hamiltonian-based formulation [Zhong, Y. D., Dey, B., & Chakraborty, A. (2019). Symplectic ODE-Net: Learning Hamiltonian Dynamics with Control]

The study by Roehrl et al. (2020) develops a neural-network-based *modeling* approach for physical systems and discusses the application of such approaches to a cart-pole model. The main similarity between the

study by Roehrl et al. (2020) and our work is that automatic differentiation and universal approximators are part of both frameworks. Roehrl et al. (2020) utilize a neural-network-based semi-physical model to account for missing features in idealized physical ODE models. As described in their manuscript, they do not use a neural-network-based control approach: “ Within the experiment, trajectories were generated by *manually controlling* the cart in a random fashion.” Our work aims at controlling complex dynamical systems whose interactions are described by a network. It would be a promising direction for future work to combine the framework of Roehrl et al. (2020) with our neural-network-based control approach to learn and control the dynamics of partially unknown systems. In the revised manuscript, we will discuss the connection of our manuscript to all of the above and several related references.

The works of Lutter et al. (2019) and Zhong et al. (2019) discuss applications of physics-informed neural networks to control theory. In contrast to those works, we show that it is possible to obtain nearly optimal control signals (*i.e.*, those that minimize the control energy) for analytically intractable, high-dimensional dynamical systems by only minimizing the distance between reached and target states.

Different neural-network-based control approaches are based on different loss functions and control methods.

- In some control frameworks, neural networks are used to obtain an approximate solution to the value function of the HJB equation [Miller, Sutton, & Werbos (1990) “Neural networks for control. The MIT Press.”], [Abu-Khalaf et al. (2005) “Nearly optimal control laws for nonlinear systems with saturating actuators using a neural network HJB approach”].
- An alternative approach is to combine Pontryagin’s maximum principle (PMP), which explicitly accounts for a control energy loss functional, with differentiable programming: [Jin, Wanxin, et al. (2020) “Pontryagin differentiable programming: An end-to-end learning and control framework.”]. In addition to relying on an energy regularization term in the cost function, the control framework of Jin et al. (2020) is based on an extension of the standard PMP equations with higher-order derivatives that require the underlying dynamical systems to be twice-differentiable. The differentiable programming framework has been applied to systems with a maximum of 13 state variables (6-DoF maneuvering quadrotor), almost two orders of magnitude smaller than some of the high-dimensional dynamical systems that we study in our manuscript.

Contrary to the above approaches, **in this work for the first time, we show that it is possible to minimize the control energy without evaluating an energy cost functional and determine nearly optimal control signals for high-dimensional complex systems** [Liu, Yang-Yu, Jean-Jacques Slotine, and Albert-László Barabási. “Controllability of complex networks.” *Nature* 473.7346 (2011): 167-173.]. Using analytical and numerical arguments, we show that AI Pontryagin is able to closely resemble the control energy of optimal control through an implicit energy regularization, resulting from the interplay of ANN initialization and an induced gradient descent. As suggested by reviewer #3, we provide a more detailed discussion of the work of Jin et al. (2020) in our revised manuscript.

For a comparison between PMP approaches, that form the basis of the work of Jin et al. (2020), and AI Pontryagin, please also see our response to comment (3) of reviewer #3.

Actions taken:

- In the revised introduction, we clarify the contributions and extend our discussion of related works: “The solution of general optimal control problems is based on two main approaches: (i) Pontryagin’s maximum principle [12, 10] (necessary condition), which is a boundary-value problem in a Hamiltonian framework, or (ii) solving the Hamilton–Jacobi–Bellman (HJB) partial-differential equation (necessary and sufficient condition) [17]. Since the HJB equation usually does not admit smooth solutions [5], different approximate dynamic programming methods have been developed [1, 2].

To extend the above approaches (i) and (ii) to complex and analytically intractable systems, different methods relying on artificial neural networks (ANNs) have been used to represent certain functions appearing in the formulation of the corresponding control problems. One possibility is to use ANNs to obtain an approximate solution to the value function of the HJB equation [11, 1]. An alternative approach is based on the solution of Pontryagin’s maximum principle via differentiable programming [6]. Control approaches that rely on Pontryagin’s maximum principle explicitly account for a control energy term in the loss function and are based on the solutions of a system’s evolution and co-state equations. In addition to relying on an energy regularization term in the loss function, the control framework of [6] is based on an extension of the maximum principle that includes higher-order derivatives, requiring the underlying dynamical systems to be twice-differentiable. Differentiable programming has been applied to control systems with a maximum of 13 state variables [6], almost two orders of magnitude smaller than some of the high-dimensional dynamical systems we study in this work.

Recent advances in automatic differentiation and physics-informed artificial neural networks also contributed to the further development of modeling and control approaches. Physics-informed neural networks use Lagrangian- and Hamiltonian-based formulations of physical models as priors for different learning tasks [13, 9, 16]. They are useful tools to model partially unknown systems [13] and have been also applied to control tasks [9, 16].

Contrary to the above approaches, we show that it is possible to generate control signals that resemble those of optimal control [15] without relying on and solving maximum principle or HJB equations. To do so, we present AI Pontryagin, an ANN that overcomes several limitations of traditional optimal control methods resulting from the analytical and computational intractability of many complex and high-dimensional control tasks.”

- We discuss the possibility of combining the study of Roehrl et al. (2020) with neural-network-based control approaches and AI Pontryagin in the revised discussion: “For such complex control tasks, it may be useful to combine physics-informed neural networks, such as those studied in [13], with our neural-network-based control approach to learn and control the dynamics of partially unknown systems.”

Reviewer #3: (2) The authors state “AI Pontryagin is able to achieve control energies similar to those of OC also for nonlinear networked dynamics”. I suspect that this statement is always true because the AGM is based on the cost function shown in Eq. (16), where the energy regularization term is introduced with a hyperparameter β . The energy constraint in the AGM can be adjusted with the hyperparameter; however, the authors state that the AI Pontryagin is similar to those of OC (trained by the AGM). Does this mean that the energy regularization of the AI Pontryagin depends on hyperparameters?

Authors: We thank reviewer #3 for this comment! AI Pontryagin **is not using any explicit energy regularization hyperparameter**. The performance of AI Pontryagin and its induced-gradient descent mechanism (see main text) depend on the choice of initial weights, which have to be small enough to let the ANN explore the vector field of the underlying dynamical system in an unbiased manner and approximate OC.

As pointed out by reviewer #3, the control energy term in the AGM depends on the regularization parameter β . In Fig. R7, we test the performance of the AGM and AI Pontryagin to control coupled Kuramoto oscillators on a square lattice with $N = 225$ nodes and different AGM regularization parameters β ranging from

Figure R7: **Controlling coupled oscillators with AI Pontryagin and the AGM.** We test the performance of AI Pontryagin and AGM [Eqs. (17) and (18)] to control coupled Kuramoto oscillators on a square lattice with $N = 225$ nodes and different AGM regularization parameter β ranging from 10^{-7} to 10^{-1} . The total simulation time is $T = 3$. **(a)** The order parameter $r(T)$. **(b)** The control energy $E_t(\mathbf{u})$. Dashed horizontal lines indicate AI Pontryagin (with different learning rates $\eta = 0.22, 0.27, 0.32$) and blue disks indicate AGM solutions.

10^{-7} to 10^{-1} . If the regularization parameter β is too large, the energy term and not the cost associated with a low degree of synchronization dominates, and the AGM fails to synchronize the oscillator system, leading to small order parameters and large control energies.

AI Pontryagin has no explicit energy regularization hyperparameter in the sense of β . It relies on an implicit energy regularization, resulting from the interplay of ANN initialization and an induced gradient descent (see main text). As requested by reviewer #3, we also analyze the dependence of the order parameter and control energy on different learning rates η for AI Pontryagin. We observe that AI Pontryagin achieves stable control of the oscillator system, as indicated by the order parameter value of $r(T) \approx 1$. At the same time, different learning rates may lead to different control energies $E_t(\mathbf{u})$. Small learning rates allow AI Pontryagin to explore the dynamical system in more detail, leading to smaller control energies and nearly optimal control solutions.

For our numerical experiments, we set $\beta = 10^{-7}$ in accordance with [3]. The learning rates η [Eq. (4)] and $\tilde{\eta}$ [Eq. (17)] are chosen such that the ratio of the order parameter values of both control methods is approximately 1.

We better explain these important details in the revised manuscript and SI.

Actions taken:

- We clarified in the revised manuscript that AI Pontryagin does not have an energy regularization parameter in the sense of parameter β of the AGM, and we also discuss how we selected the learning rates of both control methods: “AI Pontryagin directly learns $\hat{u}^*(t; \mathbf{w})$ based on the loss function (16) with a gradient descent in \mathbf{w} and *without* energy regularization term $\beta E[u]/2$.” and “The learning rates η [Eq. (4)] and $\tilde{\eta}$ [Eq. (17)] are chosen such that the ratio of the order parameter values of both control methods is approximately 1.”
- We also added more details on the AGM hyperparameter and the ANN weight initialization in the revised methods section: “In accordance with [3], the energy regularization parameter β of the AGM is set to 10^{-7} (see the SI for a more detailed analysis of the AGM control performance on β). Initially, we set all ANN weights to a value of 10^{-3} .”
- The corresponding hyperparameter analysis is shown in the SI (section IV).

Figure R8: **Runtime comparison between AI Pontryagin and the adjoint-gradient method.** We show the runtime needed to control a network of coupled Kuramoto oscillators with a subcritical coupling constant $K = 0.1K^*$. Blue disks indicate the runtime of AI Pontryagin (AIP) and orange squares indicate the runtime of the adjoint-gradient method (AGM). Simulations were performed on an Erdős–Rényi network $G(N, p)$ with $p = 0.3$. The number of time steps is 60 and the number of oscillators N ranges from 200 to 1,000. Reported runtimes are averaged over 10 realizations. Error bars are smaller than the markers.

Reviewer #3: (3) The authors state “the training time of AI Pontryagin is about two orders of magnitude smaller than that of the AGM.” This is an interesting advantage of the proposed approach, but it is unclear why the runtime (i.e., computational effort) is much smaller than the AGM. For the training, the authors also use the backpropagation-like approach shown in Fig. 1(c), which seems to be accompanied by a high computational effort, comparable to the adjoint gradient method in Eqs. (18) and (19). In my view, one of the claims is to ease the computational efforts to obtain the optimal solution and reduce the runtime. Thus, it will be important to more clearly explain this point.

Authors: We thank reviewer #3 for proposing to better explain the observed advantage of AI Pontryagin over the AGM (i.e., a numerical PMP solver). To solve control problems with AI Pontryagin, one needs to numerically integrate the primal system (not the adjoint system) and backpropagate gradients. Differences in the runtime performance between both control frameworks are associated with stiffness problems that may arise in the numerical solution of the coupled primal and adjoint equations [Eqs. (11), (12), and (18) in the revised manuscript] in the AGM, and in the corresponding gradient descent in the control function [Eq. (17) in the revised manuscript]. To support our claims, we show runtimes of AI Pontryagin and the AGM in Fig. R8. Relative runtime differences can be up to two orders of magnitude. Note, that both algorithms, AI Pontryagin and the AGM, are implemented in pytorch and executed on the same hardware.

To numerically solve the underlying dynamical systems, we use the Dormand–Prince (DOPRI) method with adaptive step size [4] in both control frameworks. For a network with N nodes, the AGM relies on the numerical integration of $2N$ nonlinear differential equations, describing the coupled primal and adjoint equations [Eqs. (11), (12), and (18) in the revised manuscript], at every iteration step. After solving the primal and adjoint systems, the AGM calculates a new estimate of the optimal control function according to the gradient descent (17) (see revised manuscript).

To solve control problems with AI Pontryagin, one needs to numerically integrate the primal system (not the adjoint system) and backpropagate gradients. To identify the main computational bottlenecks in the AGM, we performed a detailed runtime analysis of all code segments and found that the adjoint system

solver requires very small step sizes to resolve the interaction between the adjoint system and the gradient descent in the control functions. One possibility to further improve the performance of the AGM is to use a problem-tailored initialization of the control function. Such an approach requires knowledge on the mathematical structure of the optimal control signal which may not be possible to obtain, in particular for high-dimensional and analytically intractable control problems. In our simulations, we use a uniform random initialization with support [1, 2] to align the runtimes with those reported in [3]. We also tested other initialization protocols (*e.g.*, $u^{(0)}(t) = 1$ and $u^{(0)}(t) = t/T$) and were able to reduce the runtime for an Erdős–Rényi network with 225 nodes by a factor between 2–3. AI Pontryagin is still about 5–10 times faster, without optimizing the initial guess $u^{(0)}(t)$.

Actions taken:

- In the SI (section V), we discuss differences in runtime between AI Pontryagin and the AGM.
- We also refer to the runtime comparison section in the revised manuscript: “In the SI, we analyze the differences in runtime between AI Pontryagin and the AGM in more detail. To identify the main computational bottlenecks in the AGM, we performed a detailed runtime analysis of all code segments and found that the adjoint system solver requires very small step sizes to resolve the interaction between the adjoint system [Eq. (18)] and the gradient descent [Eq. (17)] in the control functions.”

Reviewer #3: (4) In my view, the robustness of optimal control is important when applying the proposed approach to a real-world control problem. However, the training method for control inputs relies on the derivatives (gradients) of the cost function and system’s states, which may be sensitive to noise. It would be better if the authors discuss about the noise robustness of the proposed approach to strengthen the impact of this work.

Authors: We agree with reviewer #3 that it is worthwhile to study the robustness of the backpropagation of cost function gradients with respect to different noise levels. As suggested in the above comment, we carry out additional numerical experiments with additive noise that acts as uncertainty on the observed reached state at time T [*i.e.*, the inputs in the cost function (5); see revised manuscript],

$$\hat{\mathbf{x}}(T) = \mathbf{x}(T) + \boldsymbol{\epsilon}, \tag{R1}$$

where $\mathbf{x}(T)$ denotes the unperturbed reached state and $\boldsymbol{\epsilon}$ is the vector whose elements are distributed according to a Gaussian distribution $\mathcal{N}(0, \sigma)$ with zero mean and variance σ^2 . The uncertainty associated with the observed reached state acts as a perturbation on the loss function (5) (see revised manuscript) and its gradients. If the signal to noise ratio is not large enough, gradients still carry enough information for efficient learning of control signals (see Fig. R9). If we introduce adaptive learning rates during the training process, it is possible to better control the level of target noise (see Fig. R10).

We will discuss these points in the revised manuscript and supplemental information.

Actions taken:

- In the SI (section II), we provide a discussion of the effect of noise on the control performance of AI Pontryagin.
- We also mention these robustness results in the revised manuscript: “In the SI, we also study the robustness of AI Pontryagin-based control with respect to different noise levels in the observed reached state.”

Figure R9: **Effect of noise on learning performance of AI Pontryagin with fixed learning rate.** For the two-state system that we discuss in the main text and different noise levels [$\sigma = 0$ in (a), $\sigma = 0.05$ in (b), $\sigma = 0.1$ in (c) and $\sigma = 0.5$ in (d)], we show the loss $J(\mathbf{x}(T), \mathbf{x}^*)$ [Eq. (5) in the main text] as a function of the number of training epochs. Gaussian noise with zero mean and variance σ^2 acts on the observed reached state $\hat{\mathbf{x}}(T)$ according to Eq. (R1). We set the learning rate to a value of $\eta = 0.1$ and use the Adam optimizer.

Figure R10: **Effect of noise and adaptive learning rates on learning performance of AI Pontryagin.** For the two-state system that we discuss in the main text and different noise levels [$\sigma = 0$ in **(a)**, $\sigma = 0.05$ in **(b)**, $\sigma = 0.1$ in **(c)** and $\sigma = 0.5$ in **(d)**], we show the loss $J(\mathbf{x}(T), \mathbf{x}^*)$ [Eq. (5) in the main text] as a function of the number of training epochs. Gaussian noise with zero mean and variance σ^2 acts on the observed reached state $\hat{\mathbf{x}}(T)$ according to Eq. (R1). We initially set the learning rate to a value of $\eta = 0.1$. After 200 training epochs, we set $\eta = 0.01$ as a fine-tuning mechanism. Computations were performed with the Adam optimizer.

Reviewer #3: (5) In Fig. 1(c): The update equation of $x(T)$ may be wrong. The authors should confirm it again.

Authors: We thank reviewer #3 for spotting this typo in Figure 1! Because of a typesetting issue, the symbol f representing the underlying dynamical system was missing. We corrected Figure 1.

Actions taken:

- We corrected Figure 1 in the revised manuscript.

Reviewer #3: (6) Page 3: “OC” may not be defined.

Authors: We apologize for that imprecision in the original manuscript. The term “OC” is an acronym of “optimal control”. In the revised manuscript, we defined the term “OC” after the first occurrence of “optimal control” on page 1.

Actions taken:

- In the revised manuscript, we defined the term “OC” after the first occurrence of “optimal control” on page 1.

Reviewer #3: (7) All in all, this manuscript is well written. The results supporting the present conclusions are appropriately shown. However, the lack of support for the above-raised points makes it unable to recommend publication of the manuscript in Nature Communications, at least in its present form.

Authors: We thank reviewer #3 for the positive feedback and the detailed suggestions for improvement. All above points are addressed in the revised manuscript and supplemental material.

References

- [1] Murad Abu-Khalaf and Frank L Lewis. Nearly optimal control laws for nonlinear systems with saturating actuators using a neural network hjb approach. *Automatica*, 41(5):779–791, 2005.
- [2] Richard E Bellman and Stuart E Dreyfus. *Applied Dynamic Programming*. Princeton University Press, 1962.
- [3] Umberto Biccari and Enrique Zuazua. A Stochastic Approach to the Synchronization of Coupled Oscillators. *Front. Energy Res.*, 8(115), 2020.
- [4] John R Dormand and Peter J Prince. A family of embedded Runge-Kutta formulae. *J. Comput. Appl.*, 6(1):19–26, 1980.
- [5] Halina Frankowska. Nonsmooth solutions of hamilton-jacobi-bellman equation. In *Modeling and Control of Systems*, pages 131–147. Springer-Verlag, 1989.
- [6] Wanxin Jin, Zhaoran Wang, Zhuoran Yang, and Shaoshuai Mou. Pontryagin differentiable programming: An end-to-end learning and control framework. *arXiv preprint arXiv:1912.12970*, 2019.

- [7] Paul L Krapivsky and Sidney Redner. Organization of growing random networks. *Phys. Rev. E*, 63(6):066123, 2001.
- [8] Yang-Yu Liu, Jean-Jacques Slotine, and Albert-László Barabási. Controllability of complex networks. *Nature*, 473(7346):167–173, 2011.
- [9] Michael Lutter, Christian Ritter, and Jan Peters. Deep lagrangian networks: Using physics as model prior for deep learning. *arXiv preprint arXiv:1907.04490*, 2019.
- [10] EJ McShane. The calculus of variations from the beginning through optimal control theory. *SIAM J. Control Optim.*, 27(5):916–939, 1989.
- [11] W Thomas Miller, Paul J Werbos, and Richard S Sutton. *Neural networks for control*. MIT press, 1995.
- [12] LS Pontryagin, VG Boltyanskii, RV Gamkrelidze, and EF Mishchenko. *Mathematical Theory of Optimal Processes [in Russian]*. Fizmatgiz Moscow, 1961.
- [13] Manuel A Roehrl, Thomas A Runkler, Veronika Brandtstetter, Michel Tokic, and Stefan Obermayer. Modeling system dynamics with physics-informed neural networks based on Lagrangian mechanics. *IFAC-PapersOnLine*, 53(2):9195–9200, 2020.
- [14] Per Sebastian Skardal and Alex Arenas. Control of coupled oscillator networks with application to microgrid technologies. *Sci. Adv.*, 1(7):e1500339, 2015.
- [15] Gang Yan, Jie Ren, Ying-Cheng Lai, Choy-Heng Lai, and Baowen Li. Controlling complex networks: How much energy is needed? *Phys. Rev. Lett.*, 108(21):218703, 2012.
- [16] Yaofeng Desmond Zhong, Biswadip Dey, and Amit Chakraborty. Symplectic ode-net: Learning hamiltonian dynamics with control. *arXiv preprint arXiv:1909.12077*, 2019.
- [17] XY Zhou. Maximum principle, dynamic programming, and their connection in deterministic control. *J. Optim. Theor. Appl.*, 65(2):363–373, 1990.

Reviewers' Comments:

Reviewer #1:

Remarks to the Author:

My previous concerns have been addressed in the revised manuscript. I thank their effort and do not have further questions.

-Gang Yan

Reviewer #3:

Remarks to the Author:

I read the revised version of the manuscript "AI Pontryagin or: How Artificial Neural Networks Learn to Control Dynamical Systems".

In the revised version, the authors have made efforts to address many comments from the reviewers, and have added the discussions and results to support their claim. The novelty and key findings of this work are clearer than the previous version. In my opinion, this paper is now acceptable for publication after addressing the following minor point:

- In Section II of SI (Noise robustness), it would be useful for readers if the authors could explain more details on the effect of adaptive learning rates for better control.

Response to Reviewers: Manuscript ID NCOMMS-21-19781A

November 21, 2021

Dear Editor and Reviewers,

Thank you again for the careful consideration of our manuscript *AI Pontryagin or: How Artificial Neural Networks Learn to Control Dynamical Systems*.

We addressed the comment of reviewer #3 in the revised SI and we also edited our manuscript to comply with the policies and formatting requirements of Nature Communications (note that we removed the colon “:” from the title in the revised manuscript and SI).

Sincerely,

Lucas Böttcher, Nino Antulov-Fantulin, and Thomas Asikis

Editor comments

Editor: Your manuscript entitled “AI Pontryagin: or How Artificial Neural Networks Learn to Control Dynamical Systems” has now been seen again by our referees, whose comments appear below. In light of their advice I am delighted to say that we are happy, in principle, to publish a suitably revised version in Nature Communications under the open access CC BY license (Creative Commons Attribution 4.0 International License).

We therefore invite you to revise your paper one last time to address the remaining concerns of our reviewers and our editorial requests in the attached document(s). At the same time we ask that you edit your manuscript to comply with our policies and formatting requirements and to maximise the accessibility and therefore the impact of your work.

Authors: We thank the editor again for the careful consideration of our manuscript “AI Pontryagin or How Artificial Neural Networks Learn to Control Dynamical Systems”. We addressed the comment of reviewer #3 in the revised SI and we also edited our manuscript to comply with the policies and formatting requirements of Nature Communications as described in the attached Author Checklist.

Point-by-point discussion

Reviewer #1

Reviewer #1: My previous concerns have been addressed in the revised manuscript. I thank their effort and do not have further questions.

Authors: We again thank reviewer #1 for the very constructive comments that helped improving this paper!

Reviewer #3

Reviewer #3: I read the revised version of the manuscript “AI Pontryagin or: How Artificial Neural Networks Learn to Control Dynamical Systems”. In the revised version, the authors have made efforts to address many comments from the reviewers, and have added the discussions and results to support their claim. The novelty and key findings of this work are clearer than the previous version. In my opinion, this paper is now acceptable for publication after addressing the following minor point:
- In Section II of SI (Noise robustness), it would be useful for readers if the authors could explain more details on the effect of adaptive learning rates for better control.

Authors: We also thank reviewer #3 again for the very helpful comments and suggestions! As pointed out by reviewer #3, adaptive learning rate schedulers can be helpful to improve the control performance of AI Pontryagin. To obtain the results that we show in Fig. S5, we reduced the initial learning rate by a factor of 10 after 200 training epochs. Other choices of learning rate schedulers (e.g., linear or exponential schedulers) are also possible. Some common learning rate schedulers are available in PyTorch¹. In the revised SI section II, we added more details on the effect of adaptive learning rates on the control performance of AI Pontryagin.

¹<https://pytorch.org/docs/stable/optim.html>

Actions taken:

- In the revised SI, we added more details on the effect of adaptive learning rates on the control performance of AI Pontryagin: “The level of observation-noise fluctuations (modeled by σ) and the gradient-descent learning rate η both affect the convergence of the learning procedure. By adaptively changing the learning rate, we reduce the impact of noise on convergence (see Fig. S5). To improve noise robustness of the training process, one can use different learning rate schedulers^a in PyTorch. A more in-depth analysis of the interplay of noise, learning rates, and stiffness of controlled differential equations is an interesting direction for future work.”

^a<https://pytorch.org/docs/stable/optim.html>